# Damage Imaging Identification of Honeycomb Sandwich Structures Based on Lamb Waves

**DOI:** 10.3390/ma16134658

**Published:** 2023-06-28

**Authors:** Chenhui Su, Wenchao Zhang, Lihua Liang, Yuhang Zhang, Qingmei Sui

**Affiliations:** 1Shandong Key Laboratory of Intelligent Buildings Technology, School of Information and Electrical Engineering, Shandong Jianzhu University, Jinan 250101, China; 2School of Control Science and Engineering, Shandong University, Jinan 250061, China

**Keywords:** honeycomb sandwich structure, Lamb Wave Tomography, damage detection, imaging

## Abstract

In the field of structural health monitoring, Lamb Wave has become one of the most widely used inspection tools due to its advantages of wide detection range and high sensitivity. In this paper, a new damage detection method for honeycomb sandwich structures based on frequency spectrum and Lamb Wave Tomography is proposed. By means of simulation and experiment, a certain number of sensors were placed on the honeycomb sandwich plate to stimulate and receive the signals in both undamaged and damaged cases. By Lamb Wave Tomography, the differences of signals before and after damage were compared, and the damage indexes were calculated. Furthermore, the probability of each sensor path containing damage was analyzed, and the damage image was finally realized. The technology does not require analysis of the complex multimode propagation properties of Lamb Wave, nor does it require understanding and modeling of the properties of materials or structures. In both simulation and experiment, the localization errors of the damage conform to the detection requirements, thus verifying that the method has certain feasibility in damage detection.

## 1. Introduction

Honeycomb sandwich structure is a special type of composite material, consisting of a composite panel on the surface and a honeycomb core in the middle. A pair of thin, strong panels are subjected to axial bending and in-plane shear loads. A thick and light honeycomb core separates the upper and lower panels to withstand the loads and transverse shear forces transferred from one panel to the other. The composite panel and honeycomb core are connected by an adhesive, which transfers shear forces to and from the honeycomb core to the panels. This is shown in Figure 1.

In recent years, some scholars have also carried out important research work on auxiliary honeycomb structures. Ren et al. [1] elaborated on the relationships among structures, materials, properties and applications of auxetic metamaterials and structures. Zhang et al. [2] studied the nonlinear transient responses of auxiliary (negative Poisson’s ratio) honeycomb sandwich plate under impact dynamic loads. Nguyen et al. [3] researched the free vibration, buckling and dynamic instability behaviors of auxiliary composite sandwich panels.

Honeycomb sandwich composite structure has the advantages of good fireproof, waterproof, sound insulation, construction elasticity, high stiffness to weight ratio, high bearing capacity and so on. It is widely used in the Marine and aerospace industries [4,5,6]. However, although all the advantages of this structure, it is susceptible to invisible damage caused by external shocks or concentrated forces during production and operation. If the damage is not detected in time, it will affect the performance and cause unforeseeable risks and losses. Therefore, the damage detection of honeycomb sandwich composites is essential. Nowadays, most of the damage detection techniques that have emerged are Non-Destructive Testing (NDT), which can be used for damage detection without damaging the tested object. Most of these methods realize damage recognition through signal or image.

Damage recognition was realized by acquiring images. Liu et al. [7] proposed a shear speckle imaging system that integrates coherent fiber illumination and fiber optic imaging bundles, and evaluated aluminum honeycomb samples to demonstrate the system’s capabilities. Shearography has the disadvantages of low efficiency and complex equipment. Defect recognition based on vision uses computer vision technology to process defect images, which provides a fast, economical and stable method for defect recognition [8]. Manujesh et al. [9] used different machine learning algorithms to classify damaged and undamaged composites, in which deep learning significantly produced good accuracy. The above method requires a large number of samples, which affects the efficiency of damage detection. Reyno et al. [10] developed and validated a surface damage detection method for honeycomb sandwich aircraft panels based on 3D scanning, which can quickly and reliably measure the surface damage dimensions, including the depth, length and area of the dents. However, 3D scanning technology is susceptible to environmental factors such as light, reflection and surface texture. Rellinger et al. [11] combined eddy current, thermal imaging and laser scanning to conduct nondestructive testing of honeycomb sandwich panels based on standard repair manuals. Eddy current is used to identify the impact site, thermal imaging is responsible for detecting debonding or delamination, and laser scanning is in charge of measuring the size of surface indentations. However, the disadvantage of thermal imaging technology is that it is easily affected by ambient temperature. Zarei et al. [12] used X-ray tomography to obtain images that showed the size, shape and location of the damage. However, X-ray detection equipment is huge and harmful to human body.

In addition, damage recognition can be realized through signal. In recent years, a number of vibration-based SHM technical standards and specifications have been developed and implemented for engineering applications [13]. Seguel et al. [14] used modal strain energy (MSE) for vibration characteristics as the de-bonding sensitive damage index of composite structures, which can correctly detect, locate and roughly quantify damage. Most past studies have used time-domain Acoustic Emission (AE) features, but recent studies have shown that frequency-domain features can help achieve valuable information [15]. Sikdar et al. [16] effectively identified the location of the damage source by obtaining dispersion curves and theoretical time domain response for damage induced acoustic emission waves in honeycomb sandwich composite structures using an acoustic source localization algorithm. Li et al. [17] used AE technology to detect the evolution of damage in honeycomb sandwich shells. However, AE technology is very sensitive to the material properties of test objects and is influenced by various external factors. Liu et al. [18] carried out damage identification on cracks of composite sandwich structures based on Gaussian process through numerical simulation, while features extracted using discrete wavelet transforms were used to train and test Gaussian models. Yin et al. [19] proposed the Basic Probability Assignment function (BPA) based on Dempster-Shafer’s evidence theory as a damage index, and searched the damage characteristics under each order of strain mode shapes, so as to identify the existence and location of disbonding defects. Yuan et al. [20] proposed a nondestructive testing method based on the band gap in honeycomb structures to detect debonding defects. The compact PZT can be surface-bonded to the structure being tested and can even be embedded between layers of composite material so that measurements can be made in situ [21]. In recent years, Electro-Mechanical Impedance (EMI) method has received special attention in defect detection of composite structures [22]. Na [23] used EMI technology to permanently connect the PZT transducer to the host structure for monitoring, and used EMI technology to create a field no-reference NDT method for detecting composite structure debonding. Chertishchev [24] demonstrated that the depth of defects in honeycomb structure can be effectively distinguished by the magnitude of mechanical impedance and the characteristic indication of cell walls on the C-scan of the entire object surface. However, the sensitivity of the impedance method decreases sharply with increasing skin stiffness or peel size.

Taking the above problems into consideration, a damage detection and localization method for honeycomb sandwich structures based on frequency spectrum and Lamb Wave is proposed in this paper. Through simulation and experiment, single damage and multiple damages were set respectively for honeycomb sandwich structure. The proposed method has a good accuracy for damage localization and has certain feasibility.

## 2. Lamb Wave Damage Mechanism

Structural health monitoring technology based on Lamb Wave is considered as one of the most promising detection methods due to its long distance propagation ability and high sensitivity to non-uniformity near the propagation path [25,26,27]. This technique identifies and locates structural damage by analyzing Lamb Wave response signals with information about structural defects [28,29]. The signal propagated in the plate is generated by stimulating the sensor and interacted with the damage, and other sensors are responsible for receiving the response signal. By processing the acquired signal, the health monitoring of the damage is finally realized.

Due to the complex geometry of honeycomb sandwich structure, the propagation characteristics of guided wave can be predicted by finite element analysis. Using the commercial software Abaqus/Explicit, the simulation and comparison of Lamb Wave in the undamaged, single damage and multiple damage honeycomb sandwich structure were carried out. The three-dimensional finite element model of honeycomb sandwich structure with or without damage was established respectively.

The structure consists of two outer skin panels and a hexagonal honeycomb core. The outer leather plate is made of carbon fiber composite material. By using SC8R cells with eight nodes and three degrees of freedom at each node, the hexagonal honeycomb core is built. The upper and lower skins are squares with a side length of 600 mm and a thickness of 1 mm, and the layup mode is [0°/90°]_4_. The parameters of its carbon fiber composite are shown in Table 1. The size of honeycomb core is 600 × 600 × 19 mm, which is aluminum material with density of 2730 kg/m^3^, Young’s modulus of 78 GPa, and Poisson’s ratio of 0.33. The structure of each honeycomb core is a hexagonal shape with a side length of 6 mm and a thickness of 0.04 mm, as shown in Figure 2. The skin plate is connected to the honeycomb core by tie. The size of each grid cell is 1 × 1 mm, thus improving the accuracy of the calculation. The honeycomb sandwich structure is composed of 2 × 10^6^ elements, which ensures the accuracy of simulation. The sampling frequency is 10 MHz, which ensures the accuracy of calculation.

Firstly, the propagation of Lamb Wave in the structure was simulated. The positions of PZT sensors and damage are shown in Figure 3. In this figure, one of the sensors was responsible for the excitation signal and the other sensor was responsible for the propagation of the received signal. A damage existed between the two sensors.

The advantage of using high frequency waves is that they can travel longer distances with little amplitude loss. Therefore, signal loss caused by high stiffness of honeycomb composites can be avoided [31]. Incident Lamb Wave can be generated by high frequency excitation.

Figure 4 shows the propagation of Lamb Wave on an undamaged and a damaged honeycomb sandwich panel, respectively, where the damage was simulated by a through-hole of 28 mm diameter. By comparing the two figures, it could be seen that if Lamb Wave encountered damage, part of it would turn into forward scattered wave through transmission and continue to propagate forward, while the other part would turn into backscattered wave through reflection. This was because the damage changed the propagation conditions of the Lamb Wave in the plate, which resulted in a mode transition. At the same time, the damage would cause the amplitude of the signal to decrease. Therefore, the damage index can be represented by the difference of signals before and after damage, so as to realize the localization imaging of damage.

## 3. Basic Principles of Lamb Wave Tomography

### 3.1. Lamb Wave Tomography

Lamb Wave Tomography is a method based on correlation analysis to identify damage by calculating the difference of signals before and after damage [32,33,34] without the need for parameters such as wave speed. Meanwhile, since Lamb Wave is susceptible to the dispersion effect, the algorithm does not require analysis of the complex multimode propagation characteristics of Lamb Wave, nor does it require understanding and modeling the properties of materials or structures. The technique includes signal comparison and image reconstruction. In the part of signal comparison, damage index can be obtained through signal differences before and after damage, as shown in Equation (1), where *DI* represents damage index.
(1)$$DI=1-\frac{\min(F_{u},F_{d})}{\max(F_{u},F_{d})}$$
$F_{u}$ and $F_{d}$ are divided into frequency spectrum peak of signal before and after damage.

In order to achieve accurate positioning of damage locations, the damage probabilities of all sensor paths are superimposed to calculate the probability distribution of each point (*x*, *y*) in *n* sensor paths. As shown in Equation (2).
(2)$$P(x,y)=\sum_{i=1}^{N-1}\sum_{j\neq i\neq 1}^{N}DI\cdot S_{ij}(x,y)$$
$S_{ij}(x,y)$ is called the spatial distribution function. By calculating the damage index of each sensor path, the damage location of honeycomb sandwich structure is determined by the probability imaging algorithm.

### 3.2. Damage Index Calculation Based on Frequency Spectrum

Since Lamb Wave signal has the characteristics of dispersion and multi-mode, using of frequency domain or time domain signals directly is not effective for signal analysis under the influence of noise environment, boundary reflections and other factors. By using complex signal processing methods, the structural integrity information of the wave propagation path can be extracted from the received signal [35]. As the processing of Lamb signals plays an important role in damage detection, it is necessary to study an efficient and accurate signal processing method. Time-frequency analysis is an effective method to process non-stationary signals, which can simultaneously extract the characteristic parameters of the Lamb signal in both the time and frequency domains, making it one of the most commonly used methods, so as to process guided wave signals.

Frequency spectrum analysis of a signal is the process of breaking a signal into a series of sinusoidal waves of different frequencies. Through frequency spectrum analysis, the frequency distribution and energy distribution of the signal can be obtained to better understand the characteristics and behavior of the signal.

The fundamental principle of frequency spectrum analysis is the Fourier transform. The Fourier transform can decompose a signal into a number of sine waves of the signal, each corresponding to a frequency component [36,37]. By performing the Fourier transform on the signal, a spectral representation of the signal can be obtained, which contains the frequency components of the signal and their corresponding amplitudes.

Suppose a continuous time signal is *f*(*t*), and its Fourier transform is *F*(*ω*), as shown in Equation (3).
(3)$$F(\omega )=\int_{-\infty }^{+\infty }f(t)e^{-j\omega t}dt$$
where *e*^−*jωt*^ is a complex exponential function, *ω* is the angular frequency of the signal. The inverse Fourier transform formula is shown in Equation (4).
(4)$$f(t)=\frac{1}{2\pi }\int_{-\infty }^{+\infty }F(\omega )e^{j\omega t}d\omega$$

The above is the Fourier transform of continuous signal. Since most of the signals collected in actual engineering are discrete signals, the Discrete Fourier Transform (DFT) is used in practical applications. For the discrete time signal *x*(*n*), its Fourier transform is *X*(*k*), and the positive transform is shown in Equation (5).
(5)$$X(k)=\sum_{n=0}^{N-1}x(n)e^{-j2\pi nk/N}\;\;(n=0,1,2,\ldots ,N-1)$$
*N* is the length of the signal and *k* is the discrete frequency. The inverse Discrete Fourier Transform formula is shown in Equation (6).
(6)$$x(n)=\frac{1}{N}\sum_{k=0}^{N-1}X(k)e^{j2\pi nk/N}\;\;(n=0,1,2,\ldots ,N-1)$$

The flow chart of damage location and imaging recognition of honeycomb sandwich structures based on frequency spectrum is shown in the Figure 5. The specific steps are as follows:Conduct damage detection of honeycomb sandwich structure by simulation and experiment respectively.Obtain the baseline signals and the signals after damage under the two approaches in step 1, respectively.Perform Fourier transform on the acquired signals to calculate the frequency spectrum of the signals.Calculate the DI by comparing the spectral differences of the signals before and after damage according to Equation (1).Calculate the damage probability distribution and realize the damage image reconstruction by Equation (2).

## 4. Finite Element Simulation

The effectiveness of the method is verified by finite element simulation. Since, in practice, the honeycomb sandwich structure is susceptible to damage caused by impacts and concentrated forces, two damage categories, single damage, and multi-damage, are set separately for the honeycomb sandwich structure in the simulation. Timely and accurate damage localization is achieved by Lamb Wave Tomography.

### 4.1. Single Damage Localization Imaging

In the single damage simulation experiment, the material parameters of the honeycomb sandwich structure are set in the same way as in the second part. In order to ensure the rigor of the study, 12 sensors were uniformly arranged in a circle with a diameter of 30 mm on the honeycomb sandwich structure, and the angle between each two sensors was 30°. Meanwhile, a damage with a diameter of 28 mm was set by means of a through hole at the position of coordinates (300, 375), as shown in the Figure 6. The S1 sensor was firstly stimulated at 50 kHz central frequencies to generate Lamb Wave, and other sensors received the forward scattered waves, back reflected waves and boundary reflected waves. The Lamb Wave signals were excited in clockwise order, and other sensors received the response signals.

Figure 7 shows the response signals before and after damage on two paths of sensor S1–S7 and sensor S5–S9 on a honeycomb sandwich structure with single damage. Where the blue line indicates the signal without damage and the red line indicates the signal after damage. As could be seen from the two figures, due to the damage on the path of sensor S1–S7, the signal was scattered, resulting in a smaller amplitude of the signal. On the path of sensor S5–S9, because there was no damage, the signals before and after damage were almost the same. The essence of signal amplitude reduction is the reduction of signal energy after scattering.

According to Equations (5) and (6), DFT was applied to the simulated signals before and after damage in the figure above, so as to obtain the frequency spectrum changes on the S1–S7 and S5–S9 paths, as shown in the Figure 8. From the Figure 8a, it could be observed that the frequency spectrum peak of the signal on the sensor S1-S7 path decreased substantially, and its values before and after the damage were 8.48 × 10^−18^ and 5.59 × 10^−18^, respectively. The reason for this change was that there was damage in the path, which affected the scattering of signals and led to the reduction of the amplitude of signals, thus resulting in the weakening of energy. However, as observed in the Figure 8b, the peak of the frequency spectrum before and after the damage of S5–S9 was almost unchanged with values of 1.341 × 10^−16^ and 1.342 × 10^−16^, respectively. The reason for this phenomenon was that there was no damage on the path, which had little effect on the energy damage. Therefore, the damage information on the honeycomb sandwich panel could be gained by obtaining the frequency spectrum of the signals.

According to the above methods, the spectrum values of signals before and after damage were obtained respectively, and the damage indexes were calculated according to Equation (1). Then, Equation (2) was used to obtain the location and imaging of damage. The image is shown in the Figure 9.

In the figure above, the position coordinate of the damage was (303.0, 376.7). Equation (7) is used to calculate the radial error of the damage, thus representing the location effect of the damage. With this equation, the radial error of the simulated localized single damage was calculated.
(7)$$e=\sqrt{{\left(x_{r}-x_{p}\right)}^{2}+{\left(y_{r}-y_{p}\right)}^{2}}$$
where, (*x_r_*, *y_r_*) represents the damage location of the simulation imaging, (*x_p_*, *y_p_*) represents the actual injury location. The imaging error of single damage was 3.45 mm.

### 4.2. Multiple Damage Localization Imaging

On the basis of the single damage, another damage was placed at the location (343, 225) in the same way to constitute multiple damage. The locations of the sensors and damages are shown in the Figure 10.

The signals of all sensor paths before and after damage were obtained through simulation. The comparison of signals on S1–S7 and S5–S9 paths is shown in the Figure 11. By observing the two figures, it could be found that the signal differences before and after damage were prominent, which was caused by the damage on both paths.

The frequency spectrum values of the signals of each path were obtained by Equations (5) and (6), where the spectrum images of S1–S7 and S5–S9 paths are shown in Figure 12. After observing the two images, the frequency spectrum peak values of the signals before and after damage were greatly reduced. The frequency spectrum peaks on the S1–S7 path were 8.48 × 10^−18^ and 5.27 × 10^−18^, and the spectral peaks on the S5–S9 path were 1.34 × 10^−16^ and 3.08 × 10^−18^, respectively. The reason for this result was that both paths contained damages, which led to reduction in signal energy.

The frequency spectrum of each sensor path was obtained respectively, and the damage index was calculated by Equation (1). Then, the location and imaging of multiple damages were obtained by Equation (2), as shown in the Figure 13.

In the figure above, the coordinates of the two injuries were (313.0, 375.0) and (324.0, 225.0). According to Equation (7), the radial errors of the damage location were calculated as 13 mm and 19 mm, respectively.

## 5. Experiment Research

The dimension of honeycomb sandwich structure is 600 × 600 × 21 mm in the experiment. Similar to the simulation, the skin and honeycomb core are made of carbon fiber composite and aluminum, respectively. The arrangement of sensors and single damage on the honeycomb sandwich plate was consistent with the simulation, as shown in the Figure 7 above. The 12 PZT sensors were uniformly arranged at equal angles in a circular fashion on the plate for excitation and reception of signals.

The experimental system is composed of computer, power amplifier, signal generator, oscilloscope and honeycomb board, as shown in the Figure 14. The signal generator is used to generate the excitation signal, the power amplifier is in charge of amplifying the signal, and the oscilloscope is responsible for collecting the signal.

The experimental procedure is described below. Firstly, the function generator output a signal with a central frequency of 50 kHz modulated by the Hanning window, which was amplified by an amplifier and loaded onto S1. Other sensors received the response signals and displayed them through the oscilloscope. 12 sensors were excited clockwise separately. The process occurred on undamaged and damaged honeycomb sandwich panels respectively to obtain baseline signals and damaged signals. In this case, the damage was formed by placing a mass block of 25 mm diameter and 25 mm height to affect the local strain field of the structure.

In the single damage experiment, the position of the mass block was the coordinate (300, 375). Figure 15 represents signals before and after damage on the S1–S7 and S5–S9 paths, respectively. Observing the two images, it can be found that the signal before and after the damage on the S1–S7 path had a large difference, which was due to the damage caused to the honeycomb plate by the pressure generated by the applied weights. While the signal on the S5–S9 path was almost unchanged, which was because the path did not contain damage.

Discrete Fourier Transform was applied to the two paths in the figure above respectively to obtain the frequency spectrum of the signals. As shown in the Figure 16.

The signal spectrum of all sensor paths was obtained, and the damage indexes were calculated. Finally, the location and imaging of single damage were obtained through the probability imaging algorithm, as shown in the Figure 17.

In the figure above, the position coordinate of the damage was (300.1, 375.1), and the radial error of the damage was 0.14 mm according to Equation (7).

In the same way, another mass block was placed at the position of the honeycomb plate (343, 225) to simulate multiple damage. The signals before and after damage along all sensor paths and the frequency spectrum values after signal processing were obtained, respectively. Among them, the signal plots are shown in Figure 18, and the corresponding frequency spectrum plots are shown in Figure 19.

The damage indexes were obtained through the signals’ frequency spectrum of all sensor paths. According to the damage probability distribution, the multi-damage location imaging was obtained, as shown in Figure 20.

The position coordinates of the two damages in the figure above were respectively (293.1, 376.6) and (367.2, 225.1). The radial errors of damage location were 7.08 mm and 2.42 cm, respectively.

## 6. Conclusions

In this paper, a damage localization imaging method for honeycomb sandwich structures based on frequency spectrum and Lamb Wave Tomography imaging is proposed. The damage indexes are obtained by the frequency spectrum values of baseline signals and damage signals, and the damage localization image is determined by probability imaging algorithm. The main findings of the study are as follows:Lamb Wave Tomography is a method to realize damage location through the difference between baseline signals and damage signals without the interference of factors such as understanding and modeling of material or structural characteristics, and has a high accuracy rate.In this paper, damage indexes calculation method based on frequency spectrum is proposed, which can effectively reflect the difference caused by damage and pay a way for damage probability imaging.Damage imaging of honeycomb sandwich structures is carried out by numerical simulation and experimental study, respectively. In both simulation and experiment, the localization errors of the damage conform to the detection requirements, which realizes the accurate localization of damage.

The proposed method for locating damage in honeycomb sandwich structures has a great precision, and also has a great application prospect for accurate damage assessment of panel structures. Since the location of the damage is unknown in practical applications, the location distribution of the sensors is also uncertain. This method cannot detect the damage outside the sensors, and has certain limitations, which is the key direction of future research.

## Figures and Tables

**Figure 1 materials-16-04658-f001:**
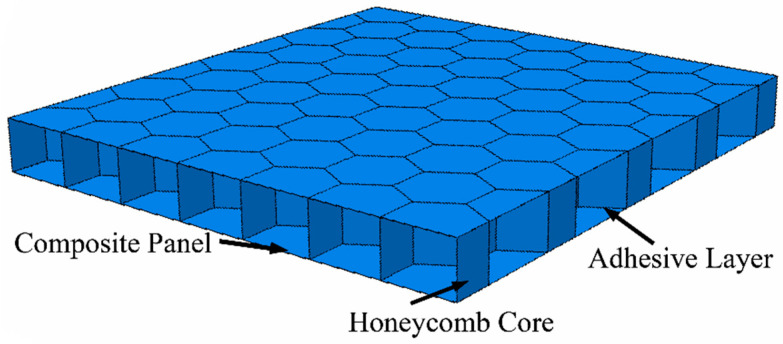
Honeycomb sandwich plate construction.

**Figure 2 materials-16-04658-f002:**
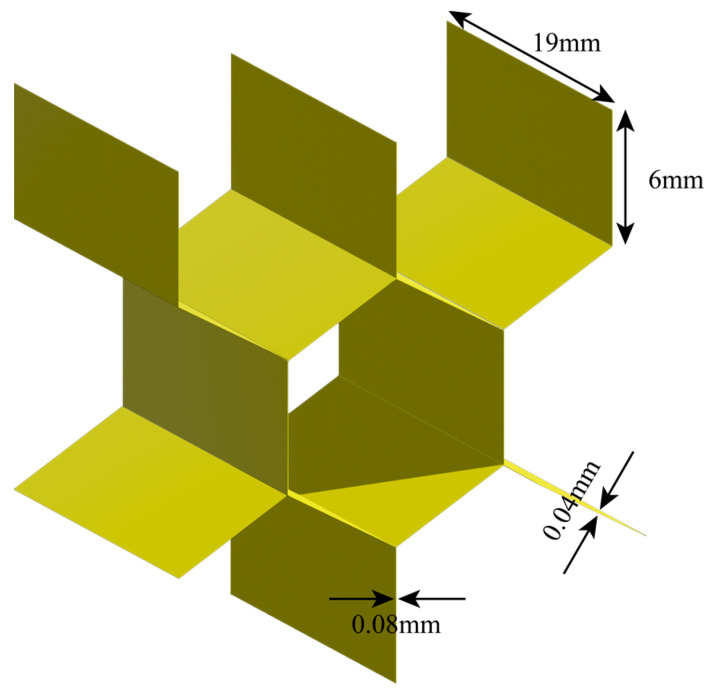
Honeycomb core structure.

**Figure 3 materials-16-04658-f003:**
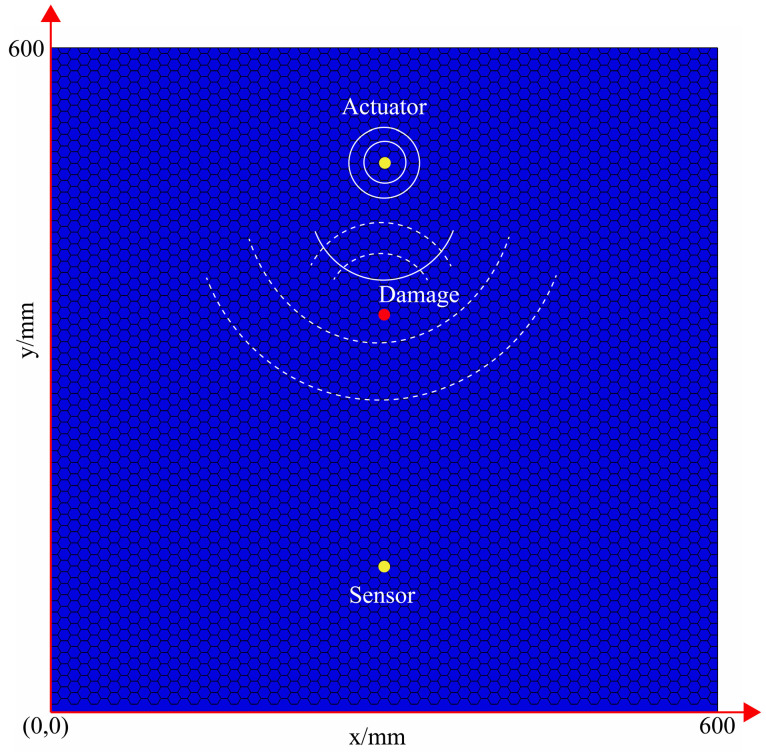
Geometric dimensions of honeycomb sandwich panels.

**Figure 4 materials-16-04658-f004:**
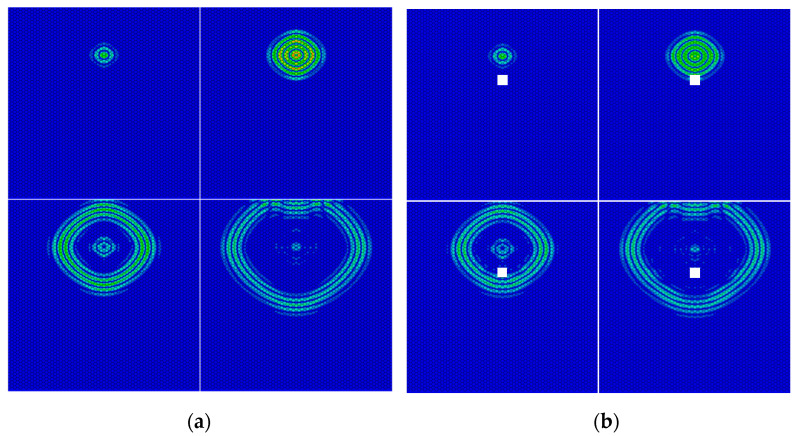
Lamb Wave propagation in honeycomb sandwich panel. (**a**) Undamaged honeycomb sandwich panel; (**b**) Damaged honeycomb sandwich panel.

**Figure 5 materials-16-04658-f005:**
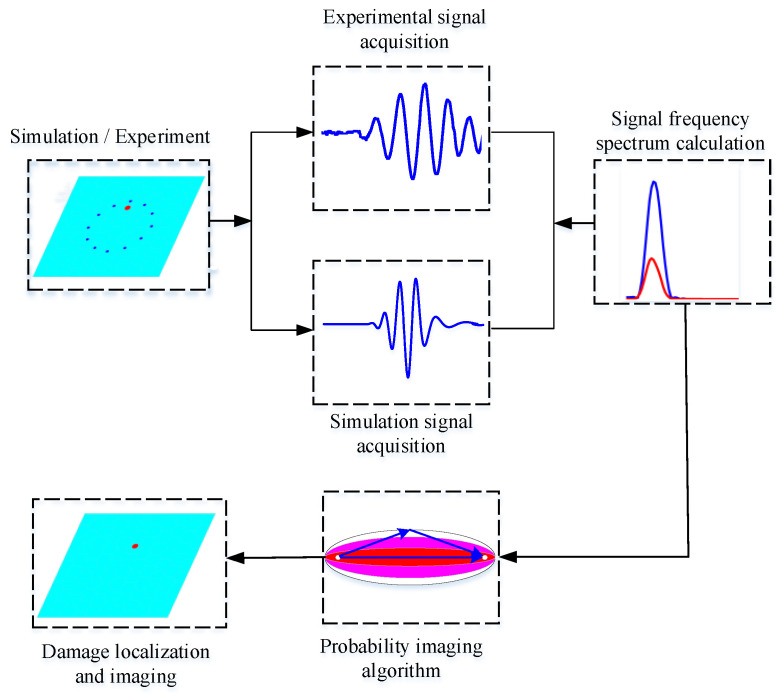
Flow chart for damage location and imaging identification of honeycomb structure.

**Figure 6 materials-16-04658-f006:**
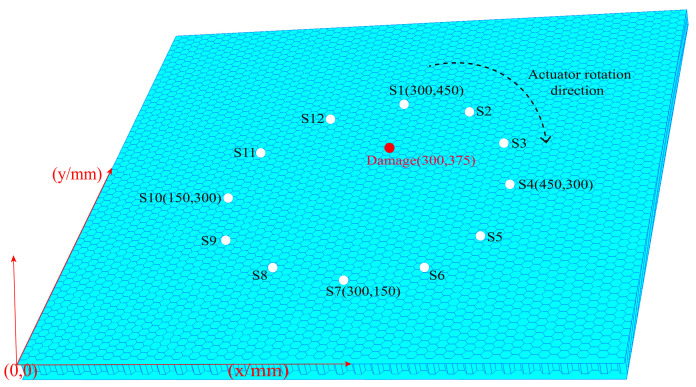
Location diagram of sensors and single damage.

**Figure 7 materials-16-04658-f007:**
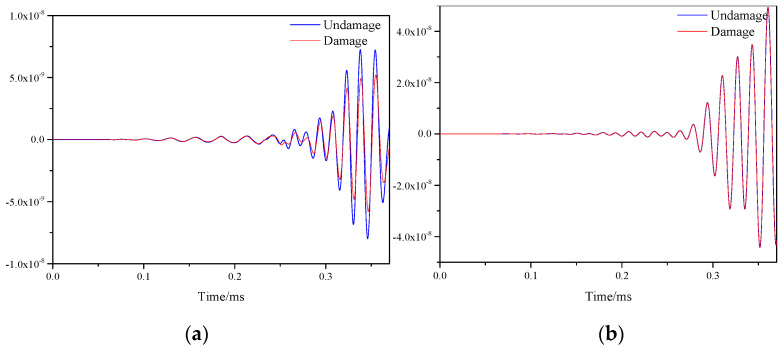
Response signals of different sensor paths before and after single damage. (**a**) S1−S7 signal sensing channel; (**b**) S5−S9 signal sensing channel.

**Figure 8 materials-16-04658-f008:**
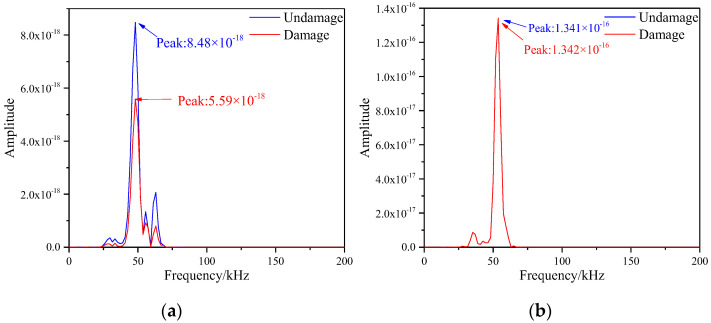
Frequency spectrum of different signal paths with single damage. (**a**) S1−S7 signal sensing channel; (**b**) S5−S9 signal sensing channel.

**Figure 9 materials-16-04658-f009:**
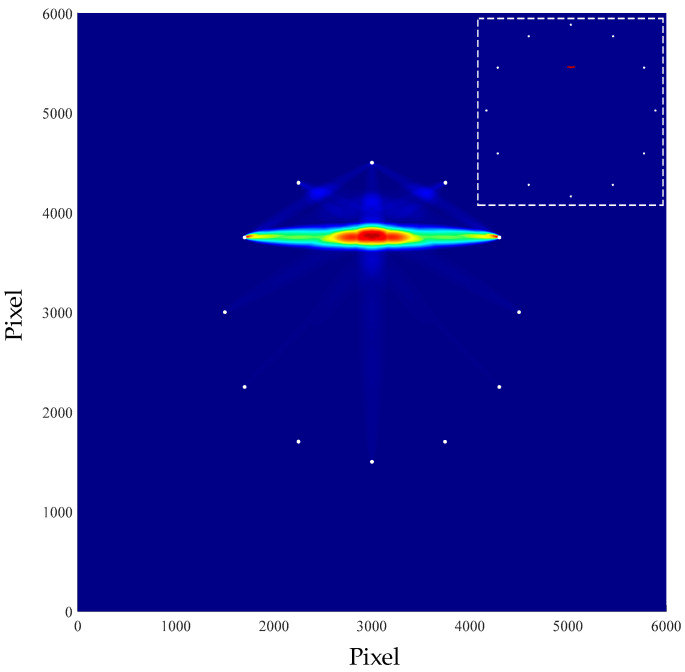
Single damage imaging result.

**Figure 10 materials-16-04658-f010:**
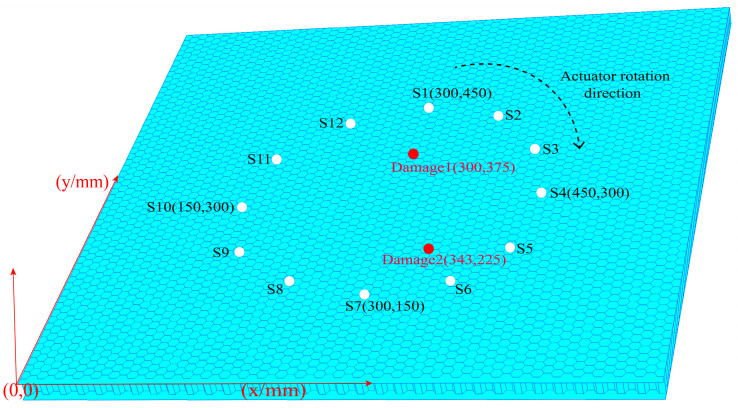
Location diagram of sensors and multiple damages.

**Figure 11 materials-16-04658-f011:**
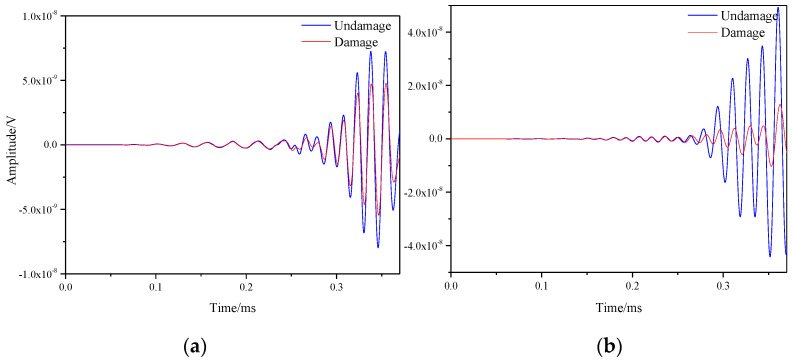
Response signals of different sensor paths before and after multiple damages. (**a**) S1−S7 signal sensing channel; (**b**) S5−S9 signal sensing channel.

**Figure 12 materials-16-04658-f012:**
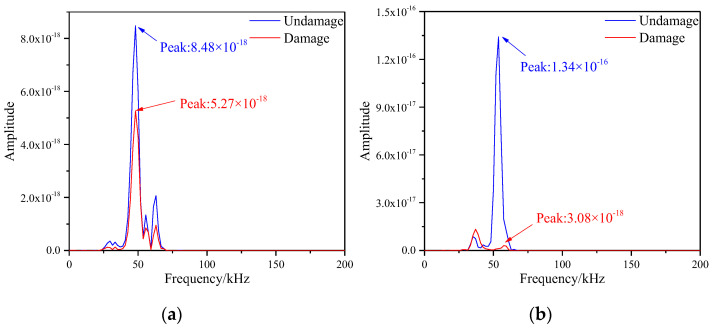
Frequency spectrum of different signal paths with multiple damages. (**a**) S1−S7 signal sensing channel; (**b**) S5−S9 signal sensing channel.

**Figure 13 materials-16-04658-f013:**
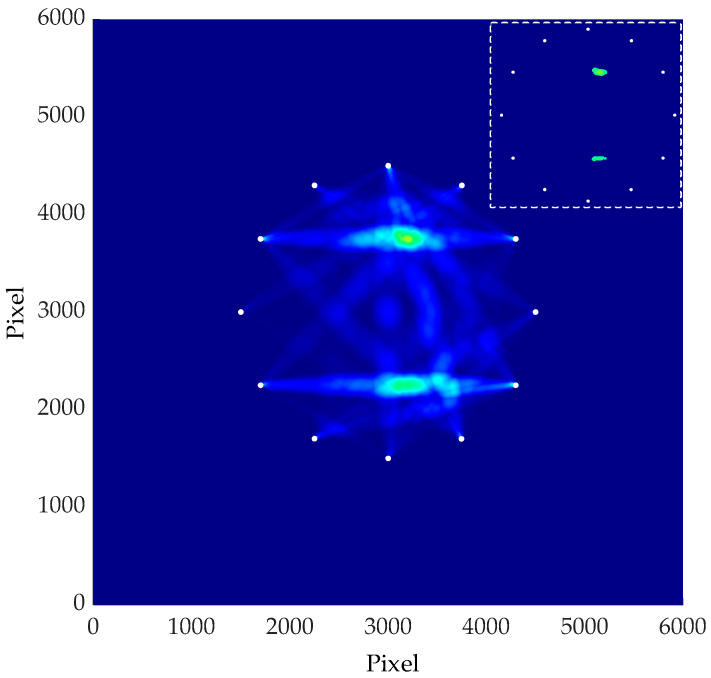
Multiple damages imaging result.

**Figure 14 materials-16-04658-f014:**
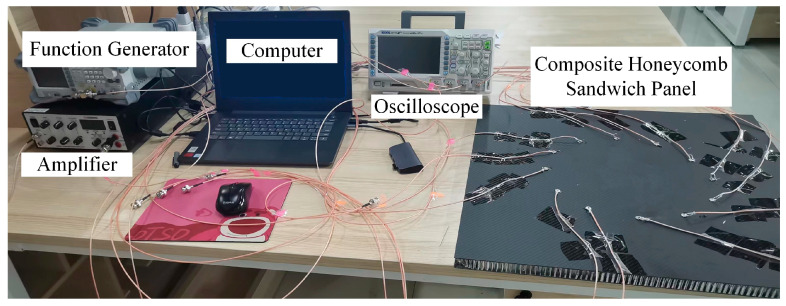
Lamb Wave Tomography imaging experiment system.

**Figure 15 materials-16-04658-f015:**
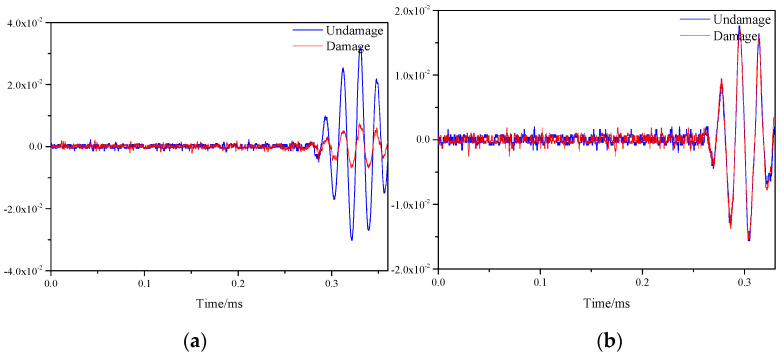
Response signals of different sensor paths before and after single damage. (**a**) S1−S7 signal sensing channel; (**b**) S5−S9 signal sensing channel.

**Figure 16 materials-16-04658-f016:**
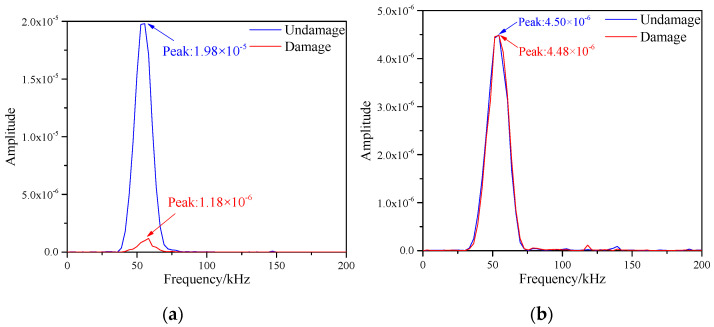
Frequency spectrum of different signal paths with single damage. (**a**) S1−S7 signal sensing channel; (**b**) S5−S9 signal sensing channel.

**Figure 17 materials-16-04658-f017:**
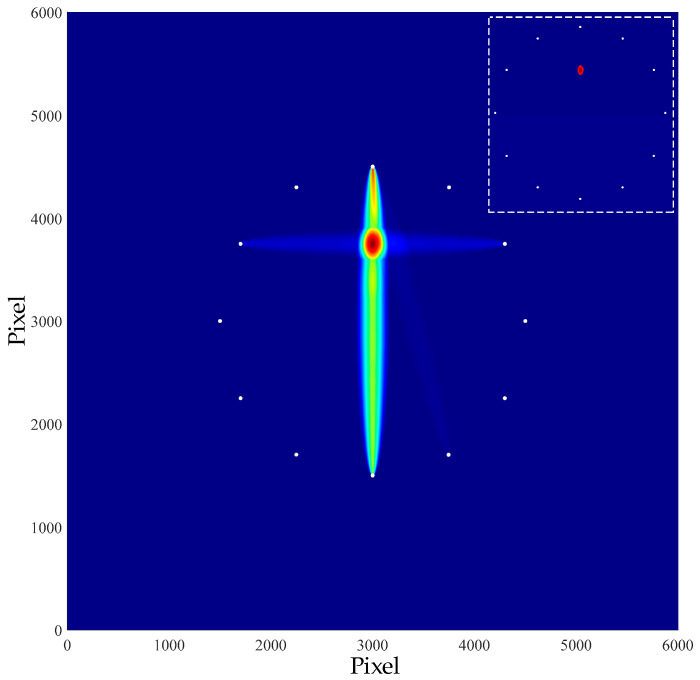
Single damage imaging result.

**Figure 18 materials-16-04658-f018:**
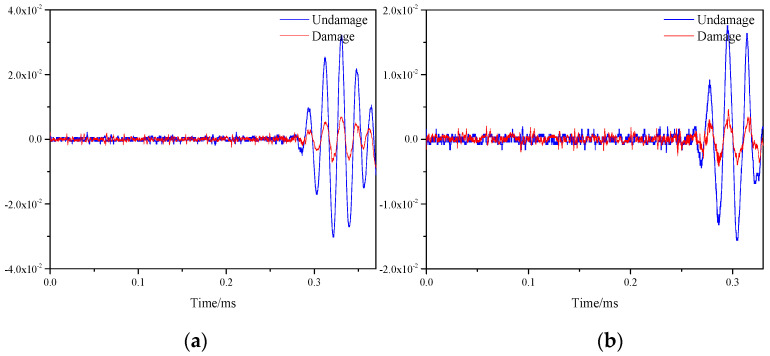
Response signals of different sensor paths before and after multiple damages. (**a**) S1−S7 signal sensing channel; (**b**) S5−S9 signal sensing channel.

**Figure 19 materials-16-04658-f019:**
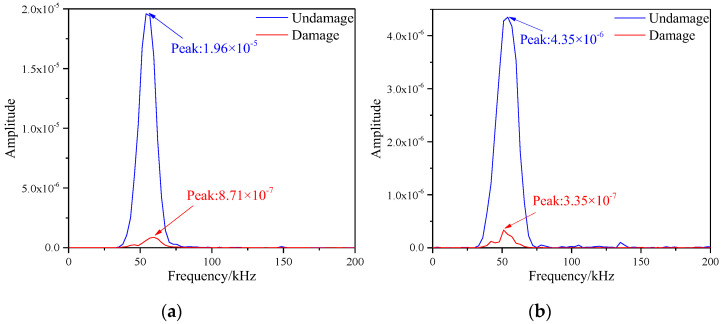
Frequency spectrum of different signal paths with multiple damages. (**a**) S1−S7 signal sensing channel; (**b**) S5−S9 signal sensing channel.

**Figure 20 materials-16-04658-f020:**
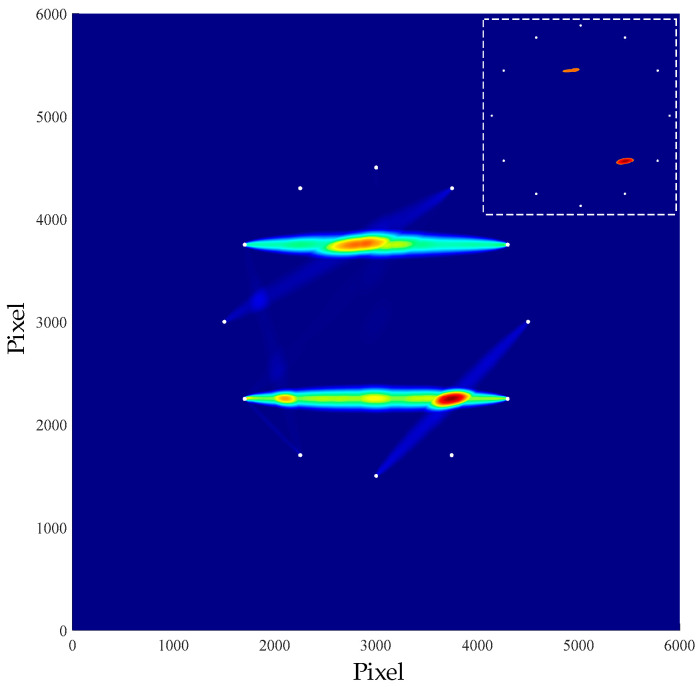
Multiple damage imaging result.

**Table 1 materials-16-04658-t001:** Parameters of carbon fiber composites [30].

Elastic Properties		Strength		Fracture Energy		Density
E_1_	110 GPa	X^T^	2093 MPa	G_ft_	10 N/mm	1700 kg/m^3^
E_2_	7.8 GPa	X^C^	870 MPa	G_fc_	10 N/mm	
ν_12_	0.32	Y^T^	50 MPa	G_mt_	1 N/mm	
G_12_	40 GPa	Y^C^	198 MPa	G_mc_	1 N/mm	
G_13_	40 GPa	S^L^	104 MPa			
G_23_	40 GPa					

## Data Availability

Data requirements can be directed to corresponding author.

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
