# Peer review of "Damage Imaging Identification of Honeycomb Sandwich Structures Based on Lamb Waves"

_materials, 2023, doi:10.3390/ma16134658_

Round 1
Reviewer 1 Report
This manuscript introduces a novel technique for detecting damage in honeycomb sandwich structures using the frequency spectrum and Lamb wave. The study includes both finite element analysis and experimental investigations. Overall, the manuscript is well-written and addresses an intriguing topic that will captivate readers. The results are meticulously presented and thoroughly discussed. The manuscript can be deemed suitable for publication after carefully addressing the following minor issues
• How to determine the optimal sensor location for detecting single and multiple damages? What is the influence of this factor on the obtained? In addition, the impact of the sensor numbers which is employed here should be investigated.
• Several necessary information regarding the finite element analysis such as the element type, mesh size, etc should be provided in the revised version.
• Some recent significant works regarding the auxetic honeycomb structures should be further discussed to enrich the literature such as Smart Mater. Struct. 27 (2018) 023001, International Journal of Impact Engineering 134 (2019) 103383, Composite Structures 259 (2021) 113213.
A language check and proofreading are necessary.
Author Response
This manuscript introduces a novel technique for detecting damage in honeycomb sandwich structures using the frequency spectrum and Lamb wave. The study includes both finite element analysis and experimental investigations. Overall, the manuscript is well-written and addresses an intriguing topic that will captivate readers. The results are meticulously presented and thoroughly discussed. The manuscript can be deemed suitable for publication after carefully addressing the following minor issues.
We appreciate the reviewer’s excellent suggestions. We thank the reviewer for the keen insight and helpful comments. All the suggestions and questions raised by the reviewer have helped us to significantly improve the quality of our manuscripts. We try our best to improve the manuscript, yellow highlight has been used to mark the modified parts. Below we discuss all the issues raised by the reviewer point by point.
Comment 1: How to determine the optimal sensor location for detecting single and multiple damages? What is the influence of this factor on the obtained? In addition, the impact of the sensor numbers which is employed here should be investigated.
Response:
In the service process of honeycomb panel, it is generally first determined by simulation analysis or engineering experience that the general area where the damage is easy to occur, and the sensors are placed in the area. According to the Lamb Wave Tomography principle, the signal indicates the damage condition on the sensor channel path, so the more sensor channels through the damage, the higher the damage location accuracy. However, considering the complexity of signal processing and damage situation, this paper used 12 sensors to verify the method.
Comment 2: Several necessary information regarding the finite element analysis such as the element type, mesh size, etc should be provided in the revised version.
Response:
In finite element analysis, the necessary information is as follows.
The structure consists of two outer skin panels and a hexagonal honeycomb core. The outer leather plate is carbon fiber composite material. By using SC8R cells with eight nodes and three degrees of freedom at each node, the hexagonal honeycomb core is built. The upper and lower skins are squares with a side length of 600 mm and a thickness of 1mm, and the layup mode is [0°/90°]4. The parameters of its carbon fiber composite are shown in Table 1. The size of honeycomb core is 600 × 600 × 19 mm, which is aluminum material with density of 2730 kg/m3, Young's modulus of 78 GPa and Poisson's ratio of 0.33. The structure of each honeycomb core is a hexagonal shape with a side length of 6mm and a thickness of 0.04 mm, as shown in Figure 2. The skin plate is connected to the honeycomb core by tie. The size of each grid cell is 1 × 1mm, thus improving the accuracy of the calculation. The honeycomb sandwich structure is composed of 2×106 elements, which ensures the accuracy of simulation. The sampling frequency is 10 MHz, which en-sures the accuracy of calculation.
Comment 3: Some recent significant works regarding the auxetic honeycomb structures should be further discussed to enrich the literature such as Smart Mater. Struct. 27 (2018) 023001, International Journal of Impact Engineering 134 (2019) 103383, Composite Structures 259 (2021) 113213.
Response:
We are sorry for missing the literature on auxiliary honeycomb structures and appreciate the literatures presented. In the revised version, the research works of some scholars about auxiliary honeycomb structure are added, and listed in the references. The specific additions are as follows.
In recent years, some scholars have also carried out important research work on auxiliary honeycomb structures. Ren et al. [1] elaborated on the relationships among structures, materials, properties and applications of auxetic metamaterials and structures. Zhang et al. [2] studied the nonlinear transient responses of auxiliary (negative Poisson's ratio) honeycomb sandwich plate under impact dynamic loads. Nguyen et al. [3] research the free vibration, buckling and dynamic instability behaviors of auxiliary composite sandwich panels.
References
- Ren, X.; Das, R.; Tran, P.; Ngo, T.D.; Xie, Y.M. Auxetic metamaterials and structures: a review. Smart Mater. Struct. 2018, 27, 023001.
- Zhang, J.H.; Zhu, X.F.; Yang, X.D.; Zhang, W. Transient nonlinear responses of an auxetic honeycomb sandwich plate under impact loads. Int J Impact Eng. 2019, 134, 103383.
- Nguyen, N.V.; Nguyen-Xuan, H.; Nguyen, T.N.; Kang, J.; Lee, J. A comprehensive analysis of auxetic honeycomb sandwich plates with graphene nanoplatelets reinforcement. Compos. Struct. 2021, 259, 113213.
Reviewer 2 Report
Major Remarks:
1. The manuscript only provides a general overview of the proposed method for damage detection in honeycomb sandwich structures using Lamb Wave Tomography. It lacks specific details about the methodology, experimental setup, performance metrics, and comparative analysis. Additionally, it does not mention any limitations or future directions for the research.
By incorporating the following additional details, the abstract would become more comprehensive, enhancing clarity, and enabling readers to better evaluate the significance and implications of the proposed method for damage detection in honeycomb sandwich structures.
1.1 Methodological Details: The abstract lacks specific information about the methodology employed in the study. It would be helpful to provide more details regarding the techniques utilized for Lamb Wave generation, signal reception, frequency spectrum analysis, and Lamb Wave Tomography.
1.2 Experimental Setup: Although the abstract mentions the use of simulations and experiments, it does not provide any specifics about the experimental setup. Including details about the materials, dimensions, and types of damage introduced in the experiments would provide important context and enable readers to better evaluate the relevance and applicability of the findings.
1.3 Performance Metrics: The abstract states that the localization errors of the damage adhere to the detection requirements, but it does not specify what those requirements are or present any quantitative metrics to assess the performance of the proposed method. Incorporating information about the specific metrics employed to evaluate the accuracy or efficiency of the damage localization would bolster the credibility and utility of the method.
1.4 Comparative Analysis: The abstract does not discuss or compare the proposed method with existing techniques or approaches in the field of structural health monitoring. Including a brief discussion or comparison with other methods would help establish the novelty and advantages of the proposed approach.
1.5 Limitations and Future Directions: It would be beneficial to acknowledge any limitations or potential areas for improvement in the proposed method. Additionally, suggesting potential future research directions or applications would provide insights into the broader scope and potential impact of the study.
2. It is noted that there is repetition of information between Section 2: Lamb Wave damage mechanism and Section 3: Basic principles of Lamb Wave Tomography in the paper. To enhance clarity and conciseness, it is important to carefully review and condense the information presented in both sections. Minimizing repetition will improve the overall readability and efficiency of conveying the research findings. It is crucial to ensure that each section provides distinct and complementary information, enabling readers to grasp the key points without encountering unnecessary duplication.
3. There appear to be issues with the structure, arrangement and flow of information in the paper. The organization of the content should be carefully reviewed and revised to improve the logical progression of the research. Reordering sections, providing clear transitions, and ensuring that each section builds upon the previous ones coherently are essential steps to enhance the overall structure and flow of the paper. By creating a well-structured and logically flowing document, readers will be able to understand and follow the research more effectively, ultimately improving the readability and comprehension of the paper.
It seems that the paper would benefit from a comprehensive English editing process to address issues related to sentence clarity and comprehension. Clear and concise language is crucial for effectively conveying research findings and ensuring that readers can easily grasp the content. Professional English editing can greatly enhance the overall quality and readability of the paper, making it more accessible and understandable to a wider audience.
Author Response
Comment 1: The manuscript only provides a general overview of the proposed method for damage detection in honeycomb sandwich structures using Lamb Wave Tomography. It lacks specific details about the methodology, experimental setup, performance metrics, and comparative analysis. Additionally, it does not mention any limitations or future directions for the research.
By incorporating the following additional details, the abstract would become more comprehensive, enhancing clarity, and enabling readers to better evaluate the significance and implications of the proposed method for damage detection in honeycomb sandwich structures.
Thanks for your comments and suggestions on our manuscript. We appreciate the reviewer for the keen insights and helpful comments. We have tried our best to improve the manuscript and made some changes in the manuscript, yellow highlight has been used to mark the modified parts. The following is the point-by-point responses to your comments.
- Methodological Details: The abstract lacks specific information about the methodology
employed in the study. It would be helpful to provide more details regarding the techniques utilized for Lamb Wave generation, signal reception, frequency spectrum analysis, and Lamb Wave Tomography.
Response:
The specific information on Lamb Wave Tomography algorithm has been added to the abstract. By Lamb Wave Tomography, the differences of signals before and after damage were compared, and the damage indexes were calculated. Furthermore, the probability of each sensor path containing damage was analyzed, and the damage image was finally realized.
Incident Lamb Wave can be generated by the following equation. The waveform in the time domain and frequency domain is shown in the figure below, and the ordinate is the normalized amplitude. The excitation frequency of the signal is 50 kHz.
(1) |
Where A represents the amplitude, fc represents the frequency, n represents the number of signal cycles (n=5), and t represents the propagation duration of the waves.
High Frequency Excitation Signal
- Experimental Setup: Although the abstract mentions the use of simulations and experiments, it does not provide any specifics about the experimental setup. Including details about the materials, dimensions, and types of damage introduced in the experiments would provide important context and enable readers to better evaluate the relevance and applicability of the findings.
Response:
In the revised version, details of the material, size and damage composition involved in the experiment have been added.
The dimension of honeycomb sandwich structure is 600 × 600 × 21 mm in the experiment. Similar to the simulation, the skin and honeycomb core are made of carbon fiber composite and aluminum, respectively. The arrangement of sensors and single damage on the honeycomb sandwich plate was consistent with the simulation, as shown in the Figure 7 above. The 12 PZT sensors were uniformly arranged at equal angles in a circular fashion on the plate for excitation and reception of signals. The damage was formed by placing a mass block of 25 mm diameter and 25 mm height to affect the local strain field of the structure.
- Performance Metrics: The abstract states that the localization errors of the damage adhere to the detection requirements, but it does not specify what those requirements are or present any quantitative metrics to assess the performance of the proposed method. Incorporating information about the specific metrics employed to evaluate the accuracy or efficiency of the damage localization would bolster the credibility and utility of the method.
Response:
Thanks for your comments and suggestions. In practical engineering applications, the error requirement of damage is generally less than 3 cm, so the error of damage location in this paper conforms to the requirement and has a certain possibility.
- Comparative Analysis: The abstract does not discuss or compare the proposed method with existing techniques or approaches in the field of structural health monitoring. Including a brief discussion or comparison with other methods would help establish the novelty and advantages of the proposed approach.
Response:
The advantages of Lamb Wave Tomography are added to the abstract. The technology does not require analysis of the complex multimode propagation properties of Lamb Wave, nor does it require understanding and modeling of the properties of materials or structures.
- Limitations and Future Directions: It would be beneficial to acknowledge any limitations or potential areas for improvement in the proposed method. Additionally, suggesting potential future research directions or applications would provide insights into the broader scope and potential impact of the study.
Response:
The methods proposed in this paper have certain limitations, which have been reflected in the conclusion section of the revised version.
Since the location of the damage is unknown in practical applications, the location distribution of the sensors is also uncertain. This method cannot detect the damage outside the sensors, and has certain limitations, which is the key direction of future research.
Comment 2: It is noted that there is repetition of information between Section 2: Lamb Wave damage mechanism and Section 3: Basic principles of Lamb Wave Tomography in the paper. To enhance clarity and conciseness, it is important to carefully review and condense the information presented in both sections. Minimizing repetition will improve the overall readability and efficiency of conveying the research findings. It is crucial to ensure that each section provides distinct and complementary information, enabling readers to grasp the key points without encountering unnecessary duplication.
Response:
In the revised version, we have deleted the damage monitoring experimental system diagram and made some modifications.
Comment 3: There appear to be issues with the structure, arrangement and flow of information in the paper. The organization of the content should be carefully reviewed and revised to improve the logical progression of the research. Reordering sections, providing clear transitions, and ensuring that each section builds upon the previous ones coherently are essential steps to enhance the overall structure and flow of the paper. By creating a well-structured and logically flowing document, readers will be able to understand and follow the research more effectively, ultimately improving the readability and comprehension of the paper.
Response:
Thank you for your comments on the organization of the paper. According to your suggestions, we have sorted out the structure of the paper, which makes the overall logic of the paper more clear.
Author Response
Thank you to the experts for reviewing the manuscript
Reviewer 4 Report
The paper is well written. Consideration of measurement or modelling errors could enhance the usefulness of results in real industrial environment. A comment related to possible nonlinear effects, like contact, could help, since in this case tools must be different.
Author Response
The paper is well written. Consideration of measurement or modelling errors could enhance the usefulness of results in real industrial environment. A comment related to possible nonlinear effects, like contact, could help, since in this case tools must be different.
Response:
Thank you for your suggestions and recognition of this manuscript. We will further analyze and study this work in the later stage.
Round 2
Reviewer 2 Report
The authors have well revised and enhanced the quality of the article.